# Shared Strength: Protective Roles of Community Resilience and Social Support in Ukrainian Forced Migration

**DOI:** 10.3390/bs15101298

**Published:** 2025-09-23

**Authors:** Martina Olcese, Paola Cardinali, Lorenzo Antichi, Francesco Madera, Laura Migliorini

**Affiliations:** 1Department of Educational Science, University of Genoa, 16128 Genoa, Italy; francesco.madera@edu.unige.it (F.M.); laura.migliorini@unige.it (L.M.); 2Department of Human and Social Science, Mercatorum University, 00186 Rome, Italy; paola.cardinali@unimercatorum.it; 3Department of Pyschology, Catholic University of Sacred Heart, 20123 Milan, Italy; lorenzo.antichi@hotmail.it

**Keywords:** refugee distress, community resilience, social support, well-being, forced migration, Ukrainian refugees

## Abstract

Forced migration following the outbreak of war in Ukraine has severely affected the psychological well-being of refugees. The community and its resources play an important role in helping refugees cope with their challenges. This study examines the role of community resilience as a mediator between refugee distress, social support and subjective well-being among Ukrainian refugees in Italy. A study was conducted with 180 Ukrainian refugees. Participants were given an online questionnaire that assessed distress, community resilience, social support and subjective well-being. A mediation analysis was conducted to test the hypothesized relationships. There was a direct negative relationship between refugee distress and well-being, which was partially mediated by community resilience. In addition, community resilience fully mediated the positive relationship between social support and well-being. High levels of distress were associated with lower levels of community resilience, which in turn predicted lower levels of well-being. Community resilience emerges as an important factor in mitigating the negative effects of refugee distress and enhancing the positive effects of social support on well-being. These findings highlight the importance of community-based psychosocial interventions aimed at promoting resilience to support the well-being and integration of refugees.

## 1. Introduction

Human mobility is a defining phenomenon of the 21st century, driven by complex interactions between economic, political, social and environmental factors ([12]). Among its most dramatic forms, forced migration, particularly that caused by armed conflict, poses profound challenges for both individuals and host communities ([37]). Since the large-scale Russian invasion of Ukraine in February 2022, Europe has seen the largest movement of refugees since the Second World War, with over eight million displaced persons, mainly women and children ([55]; [41]). The largest numbers of refugees were received by countries bordering Ukraine, especially Poland, which had nearly one million registered Ukrainian refugees by April 2024 ([55]). In comparison, Italy, though more distant, had welcomed around 175,000 refugees by May 2024, largely through emergency procedures and spontaneous hospitality from private citizens ([5]). This unprecedented humanitarian emergency has not only tested the response capacity of institutional and civil actors but has also reignited interest in psychosocial processes that promote adaptation and well-being during integration ([8]). The exceptional nature of this response was also shaped by a more favorable discursive framing compared to previous migration crises. As [56] ([56]) show, both media and EU institutions portrayed Ukrainian displacement in terms of solidarity and cultural proximity, which may have highlighted racial and religious similarities, fostering a faster and more consensual political reaction than during the more divisive 2015 refugee crisis.

Forced migration typically involves exposure to multiple traumatic and stressful experiences, ranging from pre-migration traumas related, for example, to the experience of war, to the journey often undertaken in difficult and strenuous conditions, to post-migration stressors such as legal uncertainty, unemployment, social isolation and acculturation stress ([21]). Refugees often experience chronic distress related to the breakdown of social networks, prolonged uncertainty and loss of autonomy ([2]; [20]). These stressors can lead to distress and then culminate in psychological disorders, which refer to emotional distress resulting from an inability to cope with the overwhelming demands of life and are associated with a high risk of depression, anxiety and somatic symptoms ([52]; [54]). Women, particularly those traveling alone or with children, are particularly at risk of distress due to gender vulnerabilities and care responsibilities ([53]). With this concept, the authors refer to the heightened risks that women may face in contexts of forced migration, such as exposure to gender-based violence, discrimination, exploitation, and the excessive burden of caregiving.

Despite the adversities inherent in forced displacement, various psychosocial and community resources can mitigate the effects of displacement and promote adaptation.

Among these, social support consistently emerges as a protective factor that mitigates the impact of stress and trauma and improves well-being ([24]; [13]). Social support can be emotional, material or informational in nature and can come from formal networks (institutions, services) or informal networks (family, friends, community groups). As defined in the literature, social support refers to the perception or experience of being cared for, valued and integrated into a network of mutual help, which is fundamental for emotional regulation, the ability to cope with difficulties and recovery ([57]). For refugees, access to supportive relationships, particularly those characterized by trust, empathy and solidarity, has been linked to greater life satisfaction, sense of security and resilience ([14]; [26]). In particular, social support has also been identified as a determinant of well-being, both directly, by promoting emotional stability and relational satisfaction, and indirectly, by mitigating the effects of stress and improving engagement in meaningful activities ([61]).

In recent years, research has increasingly conceptualized well-being as a multidimensional and multilevel construct, encompassing not only individual emotional functioning, but also interpersonal relationships and community engagement ([43]). Through an ecological approach, well-being can be conceptualized as relational in nature, as something that emerges from interactions between people, places and cultural values, in which social connections and social capital play a fundamental role ([3]). In migration contexts, well-being can be determined by opportunities for meaningful community participation and the social networks available for support ([18]).

In this context, community resilience offers a promising perspective for understanding how collective factors interact with individual experiences to promote well-being. Community resilience refers to the ability of a community to cope with, resist, adapt to and recover from adversity, thereby improving its functioning ([32]; [39]). It is rooted in a systemic and ecological paradigm, which shifts the focus from the individual to the collective, to social capital and shared resources to address the difficulties migrants face ([4]). In contexts of forced migration, community resilience plays a dual role: it can emerge both from ethnic networks and from the host community through inclusive policies, social trust, and opportunities for belonging ([48]). In general, it can be said that community resilience is a dynamic process promoted by different factors and therefore to be understood as a dynamic multidimensional construct ([29]). The literature classifies these factors into macro-categories, including communication and information, social support, resources and community competence ([38]).

The concept of “community”, central to the study of community resilience, is multifaceted and dynamic. Community can be defined in geographical, relational or symbolic terms ([33]) and, in migration contexts, often takes on hybrid meanings, referring simultaneously to the origin and the host country and the relational networks that emerge during different stages of migration ([51]).

Although research ([31]) has highlighted the potential of community resilience in mediating the effects of trauma and promoting recovery, empirical studies remain limited. As highlighted in a recent exploratory review ([34]), the interaction between distress, social support, community resilience and well-being among refugee populations has rarely been examined using quantitative methods, and in particular, no studies are highlighting the mediating role of community resilience in the relationship between distress and well-being and between social support and well-being.

This study aims to fill these gaps by quantitatively examining a theoretical model in which the perception of community resilience mediates the relationship between (a) social support and well-being, and (b) refugee distress and well-being, in a sample of Ukrainian refugees hosted in Italy.

Building on previous literature and the conceptual model presented in Figure 1, the following hypotheses were tested:
**H1.** *Social support is expected to be positively associated with subjective well-being (c_1_).*
**H2.** *Community resilience is hypothesized to mediate the relationship between social support and subjective well-being, resulting in an indirect positive effect (a_1_b).*
**H3.** *Refugee distress is expected to be negatively associated with subjective well-being (c_2_).*
**H4.** *Community resilience is also expected to mediate the relationship between refugee distress and subjective well-being, resulting in an indirect negative effect (a_2_b).*

## 2. Materials and Methods

### 2.1. Procedure

The data was collected through an online survey conducted via Google Forms, using a snowball sampling strategy to reach Ukrainian refugees who arrived in Italy after the large-scale Russian invasion of February 2022. Initial dissemination took place through refugee support organizations, which shared the survey link and QR code within their networks. This approach facilitated access to a hard-to-reach population by leveraging trusted community intermediaries to reduce potential discomfort or mistrust towards researchers. Snowball sampling was particularly appropriate given the post-emergency context and the need to respect participants’ privacy and autonomy.

Participation was voluntary and anonymous. Eligibility criteria included being a Ukrainian refugee and being at least 18 years old. Informed consent was obtained electronically before the start of the questionnaire. The survey, which took approximately 20–30 min to complete, included sociodemographic and migration-related questions, followed by four psychometric scales, presented in counterbalanced order to control for order effects.

The questionnaire was administered in Ukrainian to ensure linguistic and cultural accessibility. A rigorous forward-backward translation process ([6]) was conducted by certified interpreters, with discrepancies resolved through discussion to ensure conceptual and cultural equivalence. Data collection took place between April 2022 and May 2023.

### 2.2. Measures

#### 2.2.1. Refugee Distress

Refugee distress was assessed using the *Refugee Health Screener-15* (RHS-15; [19]), a self-report instrument specifically designed for refugee populations. The scale was developed to address the limitations of conventional Western tools by capturing complex psychological distress in culturally diverse and trauma-affected groups. It comprises 15 items: the first 13 assess symptoms experienced in the past month (e.g., “Do you feel emotionally numb, e.g., do you feel sad but cannot cry, cannot feel affectionate feelings?”); item 14 evaluates general coping ability; and item 15 is a “distress thermometer” measuring overall distress in the past week on a scale from 0 (no distress) to 10 (severe distress). Items 1–14 are rated on a 5-point Likert scale ranging from 0 (not at all) to 4 (extremely). The total score ranges from 0 to 66, with higher scores indicating greater psychological distress. The RHS-15 has been previously applied in studies involving Ukrainian refugees (e.g., [9]) and showed excellent reliability in this study (Cronbach’s α = 0.91).

#### 2.2.2. Community Resilience

Perceptions of community resilience were measured using the *Communities Advancing Resilience Toolkit Assessment Survey—Core Resilience* (CART; [40]), a 24-item scale originally developed to assess resilience in communities facing natural disasters. In this study, items were adapted to explicitly refer to the Ukrainian refugee community in Italy and to the context of large-scale Russian invasion of February 2022. For example, the item “My community can provide emergency services during a disaster” was modified to read: “My Ukrainian community can provide emergency services during a disaster, such as war.” The scale includes five dimensions: connection and caring (5 items), resources (5 items), transformative potential (6 items), disaster management (4 items), and information and communication (4 items). Responses are recorded on a 5-point Likert scale ranging from 1 (strongly disagree) to 5 (strongly agree), with total scores ranging from 24 to 120. Higher scores reflect stronger perceptions of community resilience. This instrument has been adapted in various refugee-related studies (e.g., [35]) and demonstrated good reliability in this study (Cronbach’s α = 0.86).

#### 2.2.3. Social Support

The *Multidimensional Scale of Perceived Social Support* (MSPSS; [62]) was used to assess emotional support from three relational sources: family, friends, and significant others. The scale consists of 12 items, four per subscale, with examples such as “There is a special person who is there for me when I need.” Responses are provided on a 7-point Likert scale (1 = strongly disagree; 7 = strongly agree), with higher scores indicating a stronger perception of support. The total score ranges from 12 to 84. The MSPSS has been previously used in research involving Ukrainian refugees (e.g., [15]) and exhibited good reliability in this study (Cronbach’s α = 0.87).

#### 2.2.4. Subjective Well-Being

Subjective well-being was assessed through the *Interpersonal, Community, Occupational, Physical, Psychological, and Economic Well-being Scale* (I-COPPE; [43]), a self-report measure comprising 21 items. It evaluates seven dimensions of well-being—interpersonal, community, occupational, physical, psychological, economic, and general—each assessed across three temporal perspectives (past, present, future). An example item is: “Considering what your life is like these days, which number would you choose?” Responses are given on an 11-point scale from 0 (worst possible life) to 10 (best possible life). The present well-being subscale (7 items, range 0–70) was used in this study, as the focus was on current psychological adjustment. The I-COPPE has been widely used in migration studies (e.g., [7]) and demonstrated good reliability in the present sample (Cronbach’s α = 0.89).

### 2.3. Data Analysis

An a priori power analysis was conducted using G*Power 3.1.5 ([11]) to determine the minimum sample size required for detecting medium effects (f^2^ = 0.15) with a statistical power of 0.80 and an alpha level of 0.05 (two-tailed). This estimate was informed by prior research showing moderate associations between social support, community resilience, and well-being (e.g., [59]). The minimum sample size required was 68 participants; the final sample of 180 provided sufficient power for the planned analyses. Prior to testing the model, key statistical assumptions were examined. The outcome variable (Y) was continuous, and both predictors (X_1_ = social support; X_2_ = refugee distress) and the mediator (M = perceived community resilience) were measured as continuous variables. Linearity among variables was verified using scatterplots. Homoscedasticity was assessed through residuals plots, and multicollinearity was checked using variance inflation factors (VIF). Potential outliers were identified and excluded to avoid distortion of parameter estimates. The normality of residuals was evaluated using histograms and Q-Q plots.

The mediation model was tested using structural equation modeling (SEM) in R Studio (version 4.3.2; [45]), employing the *lavaan* ([50]), *semPlot* ([10]), and *semTools* ([22]) packages. A series of regression paths were estimated among the independent variables (X_1_, X_2_), the mediator (M), and the dependent variable (Y). The model was estimated using maximum likelihood (ML). To assess the significance of indirect effects, a bootstrapping procedure with 1000 resamples was applied ([25]). An indirect effect was considered statistically significant if the 95% bias-corrected confidence interval did not include zero ([42]).

Specifically, two indirect effects were tested:(1)the indirect effect of social support (X_1_) on well-being (Y) through perceived community resilience (M)—path a_1_b;(2)the indirect effect of refugee distress (X_2_) on well-being (Y) through perceived community resilience (M)—path a_2_b.

## 3. Results

### 3.1. Descriptive Statistics

The study included 180 Ukrainian refugees in Italy (mean age = 41.6, standard deviation = 15.1), predominantly women (91.1%). Most participants were divorced/separated (37.8%) or married (33.9%), and 70% reported having children. The level of education was generally high, with 56.7% holding a university degree or higher. The length of stay in Italy varied, with the highest percentage (31.7%) having been there for 8–10 months. Most came from Kiev (42.2%), followed by Zaporizzja (21.1%) and Leopoli (12.2%). In terms of accommodation, 46.7% lived in reception centers, 30.5% with Ukrainian families, 17.2% with Italian families and 5.6% in independent rented accommodation. The mean scores were community resilience (M = 59.6, SD = 14.9), social support (M = 53.7, SD = 10.6), subjective well-being (M = 36.3, SD = 14.2), and refugee distress (M = 34.1, SD = 12.8) (see Table 1).

### 3.2. Mediation Model

The analysis showed that the direct effect of social support on well-being (c_1_) was not statistically significant (b = 0.04, SE = 0.07, 95% CI [−0.09, 0.17]), thus not supporting the first hypothesis (see Figure 2 and Table 2 for the full set of coefficients). However, social support demonstrated a significant positive indirect effect on well-being via community resilience (a_1_b: b = 0.20, SE = 0.05, 95% CI [0.10, 0.31]), supporting the second hypothesis. This indicates that the influence of social support on well-being is fully mediated by community resilience: greater social support was associated with higher resilience, which in turn predicted better well-being. On the other hand, refugee distress had a significant negative impact on community resilience (b = −0.57, SE = 0.06, 95% CI [−0.68, −0.45]), in line with the third hypothesis regarding its direct effect on well-being. Furthermore, an indirect pathway was also identified: higher distress led to lower community resilience, which subsequently predicted lower well-being (a2b: b = −0.27, SE = 0.06, 95% CI [−0.38, −0.17]), thus confirming the fourth hypothesis. More specifically, as distress levels increased, community resilience declined (b = −0.71), and this lower community resilience was associated with reduced well-being (b = 0.38). Although refugee distress had a significant direct negative effect on well-being (c_2_ = −0.57), the indirect effect through community resilience (a_2_b = −0.27) suggests partial mediation. This contrasts with the relationship between social support and well-being, which was entirely mediated by community resilience. In this case, community resilience acted as a protective factor, attenuating—but not eliminating—the negative impact of distress on well-being.

## 4. Discussion

The present study contributes to the growing body of research on the psychological impact of forced migration by examining the role of community resilience as a protective factor in the relationship between refugee distress, social support, and subjective well-being among Ukrainian refugees. Mediation analysis revealed a significant direct negative relationship between refugee distress and subjective well-being, which is consistent with previous findings on populations affected by armed conflict ([17]). The exposure to stressful and traumatic events experienced by migrants at different stages of migration appears to have a negative impact on the mental health of Ukrainian refugees. At the same time, the analysis identified a partial mediation of this relationship by community resilience. This suggests that, although distress has a negative impact, the community with its resources can help maintain higher levels of subjective well-being, thus mitigating the negative effects associated with refugee distress that can then lead to the onset of psychopathologies.

This finding highlights the importance of the ecological approach and the protective role of the community: individuals who belong to a community perceived as resilient may be better able to cope with the challenges of forced migration and to contain the impact of traumatic events on their mental health.

Community resilience emerges as a key protective factor following traumatic events, improving perceptions of safety and belonging ([28]). The perception that one’s community can provide accessible resources, accurate information and coordinated responses can positively influence well-being, even when individuals are exposed to high levels of stress. However, the results also indicate that community resilience is influenced by levels of distress. In fact, high levels of stress can reduce the community’s ability to collectively cope with difficulties, as already discussed in previous research ([23]), disrupting internal cohesion and limiting the activation of shared support structures. Therefore, resilience only partially mediates the relationship between stress and well-being. Furthermore, as argued by [58] ([58]), traumatic events can amplify pre-existing vulnerabilities and social inequalities, thereby weakening community resilience. In this context, promoting social support at the community level can play an important role in improving the community’s ability to cope with traumatic events such as war ([30]). Although our findings highlight the predominantly protective role of community resilience, recent studies have reported more ambivalent or selective effects. For example, [27] ([27]) found that during the COVID-19 pandemic, community resilience, although related to sense of community and well-being, did not directly mitigate the perceived impacts of the emergency. Similarly, [44] ([44]) pointed out that community resilience can have differential effects depending on citizens’ perspectives and their experiences of collective restrictions, highlighting the complexity of this construct. A tentative explanation for our more optimistic findings may lie in the specific context of forced migration: unlike the pandemic, where communities responded in very heterogeneous ways, the Ukrainian refugee community shared a common traumatic background and was committed to receiving support from both institutional and informal networks. This homogeneity of experience and the importance of immediate needs may have amplified the protective potential of community resilience, enabling it to operate more consistently as a buffer against distress.

The model also demonstrated that community resilience mediates the relationship between social support and well-being, highlighting the centrality of social support as a key promoter of community resilience processes. Through interpersonal bonds and social networks, social support improves a community’s ability to prepare for, withstand, and recover from adverse events, strengthening the systems that support resilience ([40]). This highlights the need to integrate social support structures into intervention strategies aimed at addressing the consequences of war and forced migration. These findings are further supported by recent qualitative studies, which, through the direct voices of Ukrainian refugee women, have highlighted how support from the host community and the activation of relational resources can facilitate the transition from resilience to empowerment ([16]). This is particularly relevant for women, who are often affected by separation from their families due to war, particularly from male partners who remain in their country of origin and are forced to fight.

Recent evidence highlights the centrality of emotional support in the early stages of forced migration, when individuals experience high emotional vulnerability and a strong need to make sense of their lives ([60]; [47]). Women tend to establish emotional support relationships more frequently than men ([49]), and this tendency may translate into stronger protective effects in times of crisis.

Furthermore, family ties, often the main source of support, play a central role in building and maintaining community resilience during emergencies ([1]). Support systems operate at different levels of social ecology (family, friends and significant others) and must be considered equally ([46]). Therefore, interventions should aim to strengthen social ties and ensure that refugees have access to adequate emotional and practical support. As highlighted by [36] ([36]), building community resilience also requires recognizing structural barriers, such as unequal access to support systems and services. Policies should therefore promote inclusive environments where social resources are distributed equitably and respond to the specific needs of vulnerable subgroups, including refugee women.

Each community is shaped by its unique cultural and social dynamics ([33]) and, for this reason, interventions must be tailored to reflect the evolving identity and needs of the Ukrainian community. Promoting community resilience in this way can not only improve short-term psychological well-being but also strengthen long-term integration processes. From a practical standpoint, these findings underscore the importance of designing community-sensitive psychosocial interventions that actively involve refugees in rebuilding local social networks and support systems.

Programs aimed at promoting community resilience should priorities participatory approaches, including peer-led initiatives and community-based psychosocial support, which have been shown to improve collective effectiveness and trust. In line with [36] ([36]), interventions should also pay attention to the risk of overlooking structural inequalities, such as differences in access to information, services or participation, and ensure that all subgroups within the refugee population, particularly women and those hosting them, receive the same support. Investing in training local actors (e.g., volunteers, host families, educators) on how to recognize and strengthen community resilience could be a strategic direction. Furthermore, promoting partnerships between institutional actors and informal community networks can help create more cohesive and responsive environments that can support well-being beyond the emergency phase. Supporting communities in this way can not only protect mental health but also lay the foundations for a more inclusive and sustainable integration process.

## 5. Conclusions

This study provides empirical evidence for the role of community resilience as a protective factor in mitigating the psychological distress experienced by Ukrainian refugees following forced displacement. The findings highlight the importance of adopting a socio-ecological framework to understand how individuals cope with traumatic experiences, pointing to the community and its resources as key to promoting well-being. Refugee distress has a significant negative influence on subjective well-being. However, the presence of a resilient community, perceived as capable of providing support, information and coordinated responses, can act as a buffer, mitigating the negative psychological impact of such distress.

Furthermore, the findings suggest that social support plays a dual role: it contributes directly to improving well-being and at the same time fosters the promotion of community resilience, which in turn promotes well-being.

These insights have significant implications for the development of psychosocial interventions and public policies. Programs to improve the well-being of refugees should go beyond the individual level and apply community-based strategies that strengthen interpersonal relationships, promote informal and formal networks, and improve access to emotional and practical resources. Particular attention should be paid to fostering an inclusive and participatory environment that involves refugees as active contributors to the community fabric rather than passive recipients of assistance. In this way, it is possible not only to protect mental health in the immediate aftermath of displacement but also to promote long-term community resilience and well-being. However, some limitations should be acknowledged. Specific socio-demographic and experiential variables that may influence the well-being and distress of refugee populations were not taken into account, such as socioeconomic status, family composition and dependents, length of refugee journey, length of stay in Italy, pre-existing health conditions, and language or employment status. Moreover, although some differences between men and women have been noted, gender was not included as a control variable in the present model. Future research could benefit from including such information, along with gender as a control variable, to provide a more comprehensive understanding of the factors that determine community resilience and well-being in contexts of forced migration.

## Figures and Tables

**Figure 1 behavsci-15-01298-f001:**
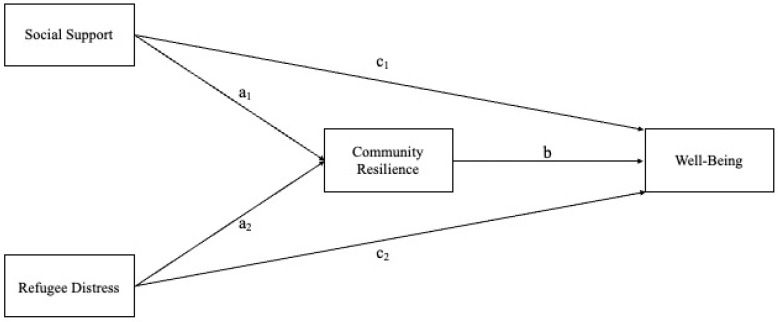
Mediation model.

**Figure 2 behavsci-15-01298-f002:**
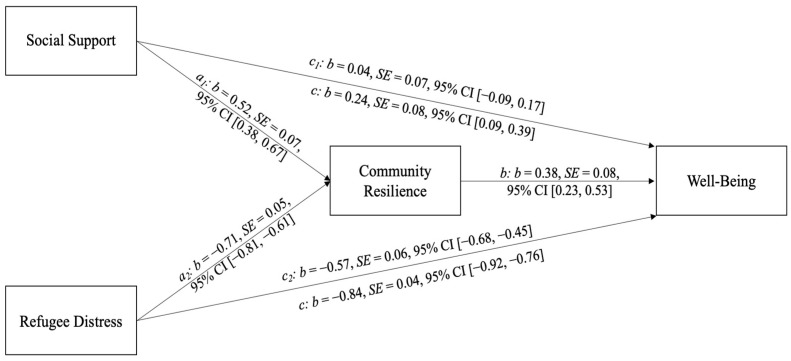
*Mediation Model Results. Note*. c = total effects; c_1_ and c_2_ = direct effects; b = unstandardized regression coefficient; SE = standard error; CI = confidence interval.

**Table 1 behavsci-15-01298-t001:** Descriptive statistics.

Characteristics	N (%)	M (SD)
Age (in years)	180	41.6 (15.1)
Gender		
Female	164 (91.1%)	
Male	16 (9.1%)	
Civil status		
Divorced/Separated	68 (37.8%)	
Married	61 (33.9%)	
Single	40 (22.2%)	
Widow	11 (6.1%)	
Having Children		
Yes	126 (70.0%)	
No	53 (29.4%)	
Educational level		
Degree/Postgraduate title	102 (56.7%)	
High school	65 (36.1%)	
Middle school	13 (7.2%)	
Length of stay		
Less than 2 months	19 (10.6%)	
From 2 to 4 months	36 (20.0%)	
From 5 to 7 months	46 (45.6%)	
From 8 to 10 months	57 (31.7%)	
More than 10 months	22 (12.2%)	
Origin Town		
Kiev	76 (42.2%)	
Zaporizzja	38 (21.1%)	
Leopoli	22 (12.2%)	
Odessa	13 (7.2%)	
Others	31 (17.2%)	
Living Arrangement		
Reception centers	84 (46.7%)	
Hosted by Ukrainian friends or relatives	55 (30.5%)	
Hosted by Italians	31 (17.2%)	
Renting a house	10 (5.6%)	
Community resilience		59.6 (14.9)
Social support		53.7 (10.6)
Subjective Well-being		36.3 (14.2)
Refugee distress		34.1 (12.8)

**Table 2 behavsci-15-01298-t002:** Mediation Model Coefficients.

	b	b*	SE	Z	*p*	95% CI
a_1_: SS → CR	0.52	0.35	0.07	7.03	<0.001	0.38, 0.67
a_2_: DISTRESS → CR	−0.71	−0.61	0.05	−13.82	<0.001	−0.81, −0.61
b: CR → WB	0.38	0.41	0.08	5.14	<0.001	0.23, 0.53
c_1_: SS → WB	0.04	0.03	0.07	0.59	0.554	−0.09, 0.17
c_2_: DISTRESS → WB	−0.57	−0.52	0.06	−9.25	<0.001	−0.68, −0.45
a_1_b: SS → CR → WB	0.20	0.14	0.05	3.73	<0.001	0.10, 0.31
a_2_b: DISTRESS → CR → WB	−0.27	−0.25	0.06	−4.89	<0.001	−0.38, −0.17
Total (c_1_ + a_1_b): SS → WB	0.24	0.17	0.08	3.14	0.002	0.09, 0.39
Total (c_2_ + a_2_b): DISTRESS → WB	−0.84	−0.77	0.04	−20.73	<0.001	−0.92, −0.76

*Note*. SS = social support; CR = community resilience; WB = well-being; b* = standardized regression coefficient; SE = standard error; CI = confidence interval.

## Data Availability

The data supporting the findings of this study are not publicly available due to ethical and privacy restrictions, as they contain sensitive information. Data may be available from the corresponding author upon reasonable request and subject to ethical approval.

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
