# Peer review of "Shared Strength: Protective Roles of Community Resilience and Social Support in Ukrainian Forced Migration"

_behavsci, 2025, doi:10.3390/bs15101298_

Round 1

Reviewer 1 Report

Comments and Suggestions for Authors

The manuscript addresses the protective role of community resources (specifically referring to resilience and social support) among Ukrainian refugees through a mediation analysis. Overall, the paper is well-written and addresses a relevant and timely topic. The literature is sound, and the discussion of the results is consistent, outlining the theoretical and practical implications stemming. However, here I detail some minor suggestions for further improvement before publication:

  • “Women, particularly those travelling alone or with children, are particularly at risk of distress due to gender vulnerabilities and care responsibilities (Tadesse et al., 2024).” (p.2, lines 58-60) – given that your sample is mainly composed by women, could you please tell something more about what you mean by “gender vulnerabilities”?
  • “Although research (Nickerson et al., 2024) has highlighted the potential of community resilience in mediating the effects of trauma and promoting recovery, empirical studies, remain limited” (p.3, lines 104-106) – I think the comma after “empirical studies” may be a typo;
  • Given that the authors make several references to some differences existing between men and women, have they considered including gender as a control variable in their model?
  • The results about the protective role of community resilience sound very interesting to me, even more in light of some recent studies rather finding a somewhat ambivalent effect of such community asset (see Mannarini et al., 2021, The potential of psychological connectedness: Mitigating the impacts of COVID-19 through sense of community and community resilience. Journal of Community Psychology, 50, 2273–2289; Procentese et al., 2023, The selective effect of lockdown experience on citizens' perspectives: A multilevel, multiple informant approach to personal and community resilience during COVID-19 pandemic. Journal of Community and Applied Social Psychology, 33(3), 719-740). Authors may want to further elaborate on this by proposing some tentative explanation or the different – and more optimistic – results they got here.

Author Response

Dear Reviewer #1,

We would like to express our sincere gratitude for taking the time to review our manuscript titled “Shared Strength: Protective Roles of Community Resilience and Social  Support in Ukrainian Forced Migration" (Manuscript ID: behavsci-3823326) submitted to  Behavioral Sciences.
We sincerely thank you for your thoughtful and constructive feedback on our manuscript. We truly appreciate the time and effort you dedicated to reading our work. Your comments have been invaluable in helping us strengthen the manuscript. Below, we address each of your points in detail. For clarity, we report each comment followed by our response and the corresponding changes made to the manuscript.

1) Women, particularly those travelling alone or with children, are particularly at risk of distress due to gender vulnerabilities and care responsibilities (Tadesse et al., 2024).” (p.2, lines 58-60) – given that your sample is mainly composed by women, could you please tell something more about what you mean by “gender vulnerabilities”?

1)We thank for this important observation. In response, we have clarified the concept of gender vulnerabilities in the manuscript (p.2, lines 60–63), where we now specify: “With this concept, the authors refer to the heightened risks that women may face in contexts of forced migration, such as exposure to gender-based violence, discrimination, exploitation, and the excessive burden of caregiving.”

2) Although research (Nickerson et al., 2024) has highlighted the potential of community resilience in mediating the effects of trauma and promoting recovery, empirical studies, remain limited” (p.3, lines 104-106) – I think the comma after “empirical studies” may be a typo.

2)We thank the reviewer for this careful observation. We confirm that the comma after “empirical studies” was a typo, and we have removed it in the revised version of the manuscript.

3) Given that the authors make several references to some differences existing between men and women, have they considered including gender as a control variable in their model?

3) We thank the reviewer for this helpful comment. In response, we have addressed this point in the limitations section of the conclusion (p.11, lines 394–402), where we now state: “However, some limitations should be acknowledged. Specific socio-demographic and experiential variables that may influence the well-being and distress of refugee populations were not taken into account, such as socioeconomic status, family composition and dependents, length of refugee journey, length of stay in Italy, pre-existing health conditions, and language or employment status. Moreover, although some differences between men and women have been noted, gender was not included as a control variable in the present model. Future research could benefit from including such information, along with gender as a control variable, to provide a more comprehensive understanding of the factors that determine community resilience and well-being in contexts of forced migration.”

4) The results about the protective role of community resilience sound very interesting to me, even more in light of some recent studies rather finding a somewhat ambivalent effect of such community asset (see Mannarini et al., 2021, The potential of psychological connectedness: Mitigating the impacts of COVID-19 through sense of community and community resilience. Journal of Community Psychology, 50, 2273–2289; Procentese et al., 2023, The selective effect of lockdown experience on citizens' perspectives: A multilevel, multiple informant approach to personal and community resilience during COVID-19 pandemic. Journal of Community and Applied Social Psychology, 33(3), 719-740). Authors may want to further elaborate on this by proposing some tentative explanation or the different – and more optimistic – results they got

4) We thank the reviewer for this insightful suggestion. In response, we have expanded the discussion in the manuscript (p.9, lines 311–324) as follows: “Although our findings highlight the predominantly protective role of community resilience, recent studies have reported more ambivalent or selective effects. For example, Mannarini et al. (2021) found that during the COVID-19 pandemic, community resilience, although related to sense of community and well-being, did not directly mitigate the perceived impacts of the emergency. Similarly, Procentese et al. (2023) pointed out that community resilience can have differential effects depending on citizens' perspectives and their experiences of collective restrictions, highlighting the complexity of this construct. A tentative explanation for our more optimistic findings may lie in the specific context of forced migration: unlike the pandemic, where communities responded in very heterogeneous ways, the Ukrainian refugee community shared a common traumatic background and was committed to receiving support from both institutional and informal networks. This homogeneity of experience and the importance of immediate needs may have amplified the protective potential of community resilience, enabling it to operate more consistently as a buffer against distress.”

Reviewer 2 Report

Comments and Suggestions for Authors

This is an excellent, interesting and compelling article that well incorporates the refugee experience into quantitative results. I have minor comments on the article itself, mostly queries that I feel would add to the strength of the findings. One thing that I feel would strengthen the article is more details on which variables or attributes were or were not significant to experiences of well-being and distress, including age of respondents, socio-economic status (if measured), age of dependents, number of dependents, duration of refugee journey, duration of time in Italy, experience of being widowed, if other adult family members (siblings, parents) were able to accompany the respondent on the journey, if they were able to speak Italian, if they had found work, if their children were in schools, pre-existing mental or physical health complaints, and -- perhaps-- a subjective measure of their general outlook/resilience (i.e., "Prior to the war, I generally saw myself as an optimistic person"). While these questions indicate my general preference for qualitative or mixed-methods work that can investigate these topics, I wonder if there might be data from your work that could shed light on these aspects and thereby strengthen your argument. 

Author Response

Dear Reviewer #2,

We would like to express our sincere gratitude for taking the time to review our manuscript titled “Shared Strength: Protective Roles of Community Resilience and Social  Support in Ukrainian Forced Migration" (Manuscript ID: behavsci-3823326) submitted to  Behavioral Sciences.
We sincerely thank you for your thoughtful and constructive feedback, which has been invaluable in helping us strengthen the manuscript.

In response to your comment, we have clarified the limitations of our study by acknowledging the absence of certain socio-demographic and experiential variables that could significantly influence well-being and distress among refugee populations. Specifically, in the revised version (p.11, lines 394–402), we now state: “However, some limitations should be acknowledged. Specific socio-demographic and experiential variables that may influence the well-being and distress of refugee populations were not taken into account, such as socioeconomic status, family composition and dependents, length of refugee journey, length of stay in Italy, pre-existing health conditions, and language or employment status. Moreover, although some differences between men and women have been noted, gender was not included as a control variable in the present model. Future research could benefit from including such information, along with gender as a control variable, to provide a more comprehensive understanding of the factors that determine community resilience and well-being in contexts of forced migration.”
